# Lipopolysaccharide-Induced Delirium-like Behaviour in a Rat Model of Chronic Cerebral Hypoperfusion Is Associated with Increased Indoleamine 2,3-Dioxygenase Expression and Endotoxin Tolerance

**DOI:** 10.3390/ijms241512248

**Published:** 2023-07-31

**Authors:** Hui Phing Ang, Suzana Makpol, Muhammad Luqman Nasaruddin, Nurul Saadah Ahmad, Jen Kit Tan, Wan Asyraf Wan Zaidi, Hashim Embong

**Affiliations:** 1Department of Emergency Medicine, Faculty of Medicine, Universiti Kebangsaan Malaysia, Jalan Yaacob Latif, Bandar Tun Razak, Cheras, Kuala Lumpur 56000, Malaysiasaadah@ppukm.ukm.edu.my (N.S.A.); 2Department of Biochemistry, Faculty of Medicine, Universiti Kebangsaan Malaysia, Jalan Yaacob Latif, Bandar Tun Razak, Cheras, Kuala Lumpur 56000, Malaysia; mlnasaruddin@ukm.edu.my (M.L.N.); jenkittan@ukm.edu.my (J.K.T.); 3Department of Medicine, Faculty of Medicine, Universiti Kebangsaan Malaysia, Jalan Yaacob Latif, Bandar Tun Razak, Cheras, Kuala Lumpur 56000, Malaysia; wan.asyraf.wan.zaidi@ppukm.ukm.edu.my

**Keywords:** delirium, kynurenine pathway, indoleamine 2,3-dioxygenase, neurodegenerative diseases

## Abstract

Indoleamine 2,3-dioxygenase (IDO) and the tryptophan–kynurenine pathway (TRP-KP) are upregulated in ageing and could be implicated in the pathogenesis of delirium. This study evaluated the role of IDO/KP in lipopolysaccharide (LPS)-induced delirium in an animal model of chronic cerebral hypoperfusion (CCH), a proposed model for delirium. CCH was induced by a permanent bilateral common carotid artery ligation (BCCAL) in Sprague Dawley rats to trigger chronic neuroinflammation-induced neurodegeneration. Eight weeks after permanent BCCAL, the rats were treated with a single systemic LPS. The rats were divided into three groups: (1) post-BCCAL rats treated with intraperitoneal (i.p.) saline, (2) post-BCCAL rats treated with i.p. LPS 100 μg/kg, and (3) sham-operated rats treated with i.p. LPS 100 μg/kg. Each group consisted of 10 male rats. To elucidate the LPS-induced delirium-like behaviour, natural and learned behaviour changes were assessed by a buried food test (BFT), open field test (OFT), and Y-maze test at 0, 24-, 48-, and 72 h after LPS treatment. Serum was collected after each session of behavioural assessment. The rats were euthanised after the last serum collection, and the hippocampi and cerebral cortex were collected. The TRP-KP neuroactive metabolites were measured in both serum and brain tissues using ELISA. Our data show that LPS treatment in CCH rats was associated with acute, transient, and fluctuated deficits in natural and learned behaviour, consistent with features of delirium. These behaviour deficits were mild compared to the sham-operated rats, which exhibited robust behaviour impairments. Additionally, heightened hippocampal IDO expression in the LPS-treated CCH rats was associated with reduced serum KP activity together with a decrease in the hippocampal quinolinic acid (QA) expression compared to the sham-operated rats, suggested for the presence of endotoxin tolerance through the immunomodulatory activity of IDO in the brain. These data provide new insight into the underlying mechanisms of delirium, and future studies should further explore the role of IDO modulation and its therapeutic potential in delirium.

## 1. Introduction

The brain is a complex organ that plays an essential role in regulating emotion and behaviour, and this adaptive process is vital in maintaining overall physical and mental health. As people age, their brain functions are naturally affected, exposing risks for age-related acute and chronic neurocognitive disorders such as delirium and dementia. It can be hard to distinguish delirium from dementia in older patients with altered sensorium, where these conditions often co-exist. In addition, most older patients diagnosed with delirium present mild symptoms [1]. While dementia is a chronic neurodegenerative condition and mainly affects cognitive function, delirium is usually acute, associated with a fluctuating deficit in neurocognitive behaviour that affects thinking, attention, and levels of consciousness [2]. Delirium affects many hospitalised older patients, and the prevalence differs according to the clinical setting. For instance, 10–31% of the elderly develop delirium in general ward admission, and the incidence can go up to 80% in intensive care units (ICUs) [3,4]. Meanwhile, the incidence of postoperative delirium (POD) is reported to be 2–50% [5]. A recent meta-analysis reported that the risk for mortality from delirium in hospitalised elderly patients was three times greater than in non-delirious patients [6]. In POD, previous reports suggest that delirium is associated with a 40% faster rate of cognitive deficit and an independent risk factor for the development of dementia [7]. Despite the substantial burden of delirium in hospital settings, many barriers exist in identifying effective detection and intervention for delirium, and the mortality from delirium has remained unchanged for the past 30 years [6].

Delirium in the elderly indicates a marker for brain vulnerability and failure of the brain to develop resilience towards an acute stimulus. There is clear evidence that elderly patients are more susceptible to delirium after surgical procedures than young patients [8]. Surgery is associated with substantial systemic inflammatory response and is significantly associated with short and long-postoperative outcomes of varying severity, including delirium [9]. On the other hand, some patients may suffer from delirium during the clinical course of mild infection. For instance, a relationship between delirium and mild illness, such as asymptomatic bacteriuria and mild COVID-19, has been observed in acute geriatric hospital admission [10,11]. In severe infections such as sepsis, the presence of delirium is one of the prognostic indicators for the probability of survival [12,13,14]. The underlying mechanism of delirium is complex, and systemic inflammation is a known precipitant for delirium. Studies suggest that several mechanisms contribute to delirium in the elderly, including neuroinflammation, reduced cerebral perfusion, increased blood–brain barrier (BBB) permeability, and neurotransmitter disturbances [15]. These age-related changes in brain morphology inevitably affect the brain’s functional reserve and increase the vulnerability of the elderly to delirium [16].

Chronic cerebral hypoperfusion (CCH) is a common ageing phenomenon and an etiological factor for neurodegenerative-related conditions such as Alzheimer’s disease and likely the relevant pathogenic link for delirium. It is established that structural alteration of the cerebral arteries induced by CCH is associated with a decrease in cerebral blood flow regulation, disrupting neuronal homeostasis [17,18,19]. Under clinical conditions, there are several causes of CCH, including small vessel ischaemia and multivessel atherosclerosis. In preclinical studies, permanent bilateral common carotid artery ligation (BCCAL) has been recognised to model an animal with CCH [20,21]. The ligation of the BCCAL for up to four weeks results in white matter injury, indicated by impaired oligodendrocyte lineage cell differentiation [22]. Meanwhile, a longer ligation of up to eight weeks is associated with age-related cerebrovascular alterations characterised by a thinned string of cerebral vessels [23]. String vessels are remnants of destroyed capillaries and lack of blood flow, which are often associated with the vascular pathologies of Alzheimer’s disease. Although the effects of CCH on the cognitive deficit are inconsistent, several animal studies reported that BCCAL-induced CCH is associated with an expression of amyloid-β and tau protein in the cerebral cortex and hippocampus [19,24]. In addition, CCH resulting from the permanent BCCAL is linked with the development of neurodegenerative changes, including impaired BBB permeability, synaptic dysfunction, altered brain-carbohydrate metabolism, neuronal loss, and oxidative stress [25,26,27]. Meanwhile, some studies reported the limitation of utilising BCCAL to model for CCH, including the variability of the outcome associated with the duration of cerebral hypoperfusion and the compensatory vascular mechanism that could not prevent neuronal death [28,29]. In support of vascular contributions to delirium, considerable evidence has reported the co-existence of CCH in delirium [30,31,32]. CCH increases the brain’s susceptibility to additional insults, including infections or metabolite imbalances, that can trigger delirium. However, the mechanism of the association between reduced cerebral hypoperfusion and delirium in the elderly is poorly understood.

With ageing, there is ‘steady’ activation of the primary brain’s innate and adaptive immune cells, such as microglia and astrocytes. These cells have the unique feature of initiating a neuroinflammatory cascade by mobilising several inflammatory profiles in the central nervous system (CNS), including inflammatory cytokines such as interleukin-1β (IL-1β), interleukin-6 (IL-6), and tumour necrosis factor-α (TNF-α), a pattern associated recognition receptors such as Toll-like receptors (TLRs), and inflammatory associated enzyme such as indoleamine 2,3-dioxygenase (IDO) [33]. The repeated increased inflammatory cytokine in the CNS is associated with microglia priming. This condition increases the susceptibility of the microglia to a secondary stimulus and exhibits an exaggerated inflammatory response. Uniquely, under the influence of cytokines, ‘primed’ microglia express heightened IDO levels, which serve immunoregulatory and tolerogenic functions [34,35,36]. Increased expression of IDO enhances amino acid tryptophan (TRP) degradation via the kynurenine pathway (KP), which can affect the host immune system and CNS. TRP is the primary precursor for brain serotonin and melatonin neurotransmitter and helps to regulate the sleep–wake cycle, attention, mood, and pain. TRP degradation via the KP produces neuroactive downstream metabolites such as kynurenine (KYN), 3-hydroxykynurenine (3-HK), kynurenic acid (KYNA), and quinolinic acid (QA). KYNA is an *N*-methyl-d-aspartate (NMDA) receptor antagonist and has been implicated in neuroprotective activity [37,38]. Meanwhile, QA is an agonist of the NMDA and has a potent neurotoxic effect. With increased IDO expression, the neuroprotective effect of KYNA could be diminished by the production of QA [39].

Under chronic exposure to immune stimuli, activation of TRP degradation via KP activity is implicated in neurocognitive consequences leading to dementia [40]. On the other hand, acute activation of IDO and altered TRP-KP in the ageing brain could contribute to behavioural and cognitive consequences such as delirium [41]. Previously, Peng and colleagues proposed sets of battery assessments in a preclinical model, suggesting that natural and learned behaviour were both impaired in POD [42]. Both natural and learned behaviours depend on the intactness of thinking, attention, and consciousness and are said to be impaired in delirium. The study indicated that changes in the buried food test (BFT), open field test (OFT), and Y-maze test were equivalent to the confusion assessment method (CAM) for the assessment of delirium in humans. Therefore, the current study had two main goals. First, we examined the effects of systemic lipopolysaccharide (LPS) administration in a rat model of chronic cerebral hypoperfusion (CCH)-induced neurodegeneration to emulate delirium in the elderly, utilising a set of the mentioned battery assessment tools. Secondly, we elucidated the role of the KP in the rat model of delirium, examining the effects of the LPS challenge on time-course changes in TRP-KP metabolites.

## 2. Results

### 2.1. Verification Phase

#### 2.1.1. The Effects of CCH on the Behaviour

To elucidate the behavioural changes related to CCH-induced neurodegeneration, BFT, OFT, and Y-maze tests were conducted four weeks (day 28) and eight weeks (day 56) after permanent bilateral common carotid artery ligation (BCCAL) surgery. At four weeks, the BCCAL rats exhibited significantly higher time spent in the centre of the open field, a higher number of arms visited, higher entries in the novel arm, and lower latency to the centre of the Y-maze when compared to the sham-operated rats, *p* < 0.05 (Figure 1). All test batteries did not significantly differ between the groups at eight weeks. However, the BCCAL rats had a significant time-dependent reduction in the total distance travelled from week 4 to week 8 after surgery (50.6% reduction, *p* < 0.05). These findings indicate that BCCAL surgery transiently impaired locomotor activity at four weeks, and there were no differences in any of the behavioural tests between groups at eight weeks after surgery.

#### 2.1.2. The Effects of CCH on the Brain Amyloid-β (Aβ) Depositions

To investigate whether CCH-induced neurodegeneration is related to amyloid-β deposition, we measured amyloid deposits in the brain eight weeks after surgery. The study observed a significant increase in the levels of Aβ-42 in the BCAAL rats compared to the sham-operated rats in the hippocampus and cerebral cortex, *p* < 0.001 (Figure 2).

### 2.2. Experimental Phase

#### 2.2.1. The Effects of LPS Treatments on the Body Weight (BW) in a Rat Model of CCH

At baseline, the % weight loss significantly differed between groups, F(2,18) = 16.134, *p* < 0.001 (one-way ANOVA). The saline and LPS-treated BCCAL rats had a significant reduction in BW compared to the sham-operated rats (*p* < 0.05) (Figure 3). The mortality rate from BCCAL surgery was 28% (20 of 70), including nine rats with significant neurological deficits and visual impairments who were excluded from LPS treatment. LPS administration robustly reduced the BW of BCCAL and sham-operated rats, with a significant difference between groups observed 24 h after LPS injection, F(2,18) = 7.451, *p* = 0.004 (one-way ANOVA). Post hoc analysis demonstrated that the mean % change in BW for the BCCAL + LPS and sham + LPS rats had significantly reduced compared to the BCCAL + saline rats, *p* = 0.010 and 0.016, respectively. There was no significant difference in the BW between groups at 48 and 72 h post LPS injection.

#### 2.2.2. The Effects of LPS Treatment on the Natural and Learned Behaviours in a Rat Model of CCH

To elucidate delirium-like behaviours associated with LPS administration, we performed a BFT and OFT to assess natural behaviour and a Y-maze test to evaluate learned behaviour. Figure 4 demonstrates the time-course effects of LPS treatment on the natural and learned behaviours in CCH rats at baseline (before LPS) and 24 h apart after LPS administration. In the BFT, there was no significant difference in the latency to eat food at baseline between experimental groups. LPS treatment caused significant changes in the latency to eat food between experimental groups at 48 h after LPS treatment, H(2) = 6.670, *p* = 0.036 (Kruskal–Wallis test). Dunn’s pairwise test demonstrated that LPS treatment in the BCCAL rats significantly increased the latency to eat food compared to the BCCAL + saline rats, with a median of 300 s vs. 150 s, *p* = 0.013. Nonetheless, the impairment was transient, and no significant difference was observed in the latency to eat food at 24 and 72 h post-LPS treatment (Figure 4A). In contrast, no significant difference was observed in the sham + LPS rats compared to the BCCAL + saline rats at all time points.

In the OFT, there was no significant difference in all baseline parameters between experimental groups. LPS administration transiently reduced the total distance travelled with significant differences between groups at 48 h, H(2) = 13.85, *p* = 0.001 (Kruskal–Wallis test), but not at 24 and 72 h post-LPS treatment. Dunn’s pairwise test demonstrated that LPS treatment in the sham-operated rats significantly reduced the total distance travelled, a median of 5 m vs. 10 m, *p* = 0.006, compared to the BCCAL + saline rats (Figure 4B1). No significant differences in the total distance travelled were observed between BCCAL + LPS rats compared to the BCCAL + saline rats. Meanwhile, LPS administration reduced the duration of time spent in the center of the open field, with a significant difference between groups observed at 24 h, H(2) = 13.286, *p* = 0.001, at 48 h, H(2) = 9.321, *p* = 0.009, and at 72 h, H(2) = 11.302, *p* = 0.004 (Kruskal–Wallis test) post-LPS treatment. Compared to the BCCAL + saline rats, the BCCAL + LPS rats had a fluctuated reduction in the duration of time spent with significant differences observed at 24 and 72 h post-LPS, median 2 s vs. 4 s, *p* = 0.046, and median 0.96 s vs. 4.23 s, *p* = 0.031, irrespectively. Meanwhile, LPS treatment in the sham-operated rats significantly reduced the duration of time spent throughout the time points compared to the BCCAL + saline rats, *p* < 0.05 (Figure 4B2). In the freezing time, there was a transient increase in the freezing time with a significant difference between groups at 48 h post-LPS treatment, H(2) = 6.372, *p* = 0.041 (Kruskal–Wallis test). Dunn’s pairwise test demonstrated that the sham + LPS rats had a significant increase in the freezing time compared to the BCCAL + saline rats, median of 291 s vs. 276 s, *p* = 0.012. In contrast, LPS treatment did not significantly affect the freezing time in the BCCAL rats (Figure 4B3). Then, we observed that LPS treatment decreased the latency to the center with a significant difference between the group at 24 h, H(2) = 11.668, *p* = 0.003, at 48 h, H(2) = 10.034, *p* = 0.007, and at 72 h, H(2) = 6.991, *p* = 0.030 (Kruskal–Wallis test). Dunn’s pairwise test demonstrated that the sham + LPS rats had a significant reduction in the latency to the centre compared to BCCAL + saline rats at all time points, *p* < 0.05. In contrast, no significant difference in the latency to the centre was observed in the BCCAL + LPS rats compared to the BCCAL + saline rats (Figure 4B4). These findings indicate that LPS treatment in the CCH rats had caused deficits in natural behaviour, and the impairment was acute, transient, and fluctuated. In contrast, the sham-operated rats exerted a robust impairment in the natural behaviour under the same dose of LPS.

Regarding learned behaviour, we found a statistically significantly difference in the number of arms visited in the Y-maze between groups at baseline, H(2) = 7.686, *p* = 0.021 (Kruskal–Wallis test). LPS treatment in the sham-operated rats significantly decreased the number of arms visited at baseline compared to the BCCAL + saline rats, *p* = 0.005. LPS administration reduced the number of arms visited with significant differences between groups at 24 h, H(2) = 13.847, *p* = 0.001, at 48 h, H(2) = 12.731, *p* = 0.002, and at 72 h, H(2) = 6.930, *p* = 0.031 (Kruskal–Wallis test). The effects of LPS in the sham-operated rats were more substantial, with a significant decrease in the number of arms visited at all time points compared to the BCCAL + saline rats, *p* < 0.05 (Dunn’s pairwise post hoc test). In contrast, the BCCAL + LPS rats exerted a significant transient deficit in the number of arms visited at 48 h compared to the BCCAL + saline rats, *p* = 0.030 (Figure 4C1). Meanwhile, there were no significant differences in the number of novel arm entries and the duration in the novel arm between experimental groups at baseline. LPS administration decreased the number of novel arm entries in both BCCAL and sham-operated rats with significant differences between groups at 24 h, H(2) = 13.284, *p* = 0.001 and at 48 h, H(2) = 11.217, *p* = 0.004 (Kruskal–Wallis test), but not at 72 h. Dunn’s pairwise test demonstrated that LPS treatment in the BCCAL rats significantly reduced the novel arm entries compared to the BCCAL + saline rats at 24 h and 48 h, *p* < 0.05. Furthermore, sham-operated rats had a significant robust reduction in the number of novel arm entries compared to the BCCAL + saline rats at both time points, *p* < 0.001 (Dunn’s pairwise post hoc test) (Figure 4C2). Similarly, under the effects of LPS, the duration in the novel arm significantly differed between the group at 24 h post-LPS, H(2) = 13.738, *p* = 0.001 (Kruskal–Wallis test), but 48 and 72 h. Dunn’s pairwise test revealed that the duration in the novel arm was significantly reduced in the sham + LPS rats compared to the BCCAL + saline rats, <0.001 (Figure 4C3). In contrast, no significant difference was observed in the BCCAL + LPS rats compared to the BCCAL + saline rats. Collectively, LPS treatment caused impairment in natural and learned behaviour, and these performance deficits depend on the intactness of thinking, attention, and consciousness. Importantly, these behaviour impairments in the CCH rats were acute, transient, and fluctuated, equivalent to delirium-like features in older persons. In contrast, the sham-operated rats exerted constant and more substantial performance deficits throughout the assessment period.

#### 2.2.3. The Effects of LPS on the KP Metabolites in a Rat Model of CCH

Figure 5 demonstrates the effects of LPS treatment on the serum TRP-KP metabolites at different time points. At baseline, the TRP levels did not significantly differ between groups. However, the sham + LPS rats had significantly lower KYN and higher KYNA levels at baseline compared to both BCCAL rats on saline and LPS irrespectively, *p* < 0.001. The sham + LPS rats also had a significantly lower baseline KYN/TRP, *p* < 0.05, and a higher baseline KYNA/QA ratio, *p* < 0.001, compared to both BCCAL on saline and LPS rats. These baseline findings indicate increased KP activity in the BCCAL rats, with evidence of increased breakdown towards the neurotoxic branch compared to the sham-operated rats. LPS administration in the BCCAL rats was briefly associated with downregulated KP, evidenced by a significantly lower KYN/TRP ratio at 48 h compared to the sham + LPS rats, *p* < 0.05. We observed no significant differences in the QA levels post LPS treatment, except at 72 h, in which the BCCAL + LPS rats had a significant increase in the QA levels compared to the BCCAL + saline rats, *p* < 0.001. Meanwhile, the sham + LPS rats exerted a persistent significant increase in the KYNA levels post LPS compared to both BCCAL on saline and LPS irrespectively, *p* < 0.001. In addition, the sham + LPS rats also had a significantly higher KYNA/TRP ratio at 72 h compared to both BCCAL rats on saline and LPS, *p* < 0.05. These findings indicate that the LPS-treated CCH rats exerted refractory KP activation in the serum compared to the saline-treated CCH rats. In contrast, LPS-treated sham-operated rats had substantial KP activation compared to both CCH on saline and LPS rats.

The levels of TRP, KYN, and KYN/TRP ratio in the hippocampus and cerebral cortex at 72 h post-LPS are shown in Figure 6. There was a significant difference in the levels of TRP in the hippocampus but not in the cerebral cortex between groups, F(2,15) = 8.146, *p* = 0.004 (one-way ANOVA). Post hoc analysis revealed that the sham + LPS rats had a significantly higher level of TRP in the hippocampus compared to the BCCAL + saline (*p* = 0.021) and BCCAL + LPS (*p* = 0.015) rats. In contrast, the hippocampal TRP levels in the BCCAL + LPS rats were not significantly different compared with BCCAL + saline rats. Meanwhile, no significant differences in the levels of KYN in the hippocampus and cerebral cortex were observed at 72 h post-LPS administration. We further determined the KYN/TRP ratio and identified that the sham + LPS rats had a significantly lower KYN/TRP ratio in the hippocampus than the BCCAL + LPS rats (*p* = 0.032). Similarly, no significant difference in the KYN/TRP ratio was observed between the BCCAL + LPS and BCCAL + saline rats. These findings indicate that at 72 h post-LPS, the sham-operated rats exerted downregulated hippocampal KP activity compared to the CCH rats and suggest the transient effects of LPS.

We further measured the expression of IDO in the hippocampus and cerebral cortex (Figure 7). One-way ANOVA revealed that the hippocampal expression of IDO in the sham + LPS rats was significantly lower as compared to the BCCAL + saline (*p* = 0.009) and BCCAL + LPS (*p* = 0.009) rats, respectively, F(2,15) = 34.671, *p* < 0.001. There was no significant difference in the expression of IDO between BCCAL rats exposed to saline and LPS (*p* = 0.184). In addition, there was no significant difference in the expression of IDO in the cerebral cortex between experimental groups.

Next, we examined the levels of neuroactive metabolites from the KP. Interestingly, we identified that the sham + LPS rats had significantly elevated levels of hippocampal QA, F(2,15) = 17.997, *p* < 0.001 (one-way ANOVA), and post hoc analysis identified the levels were significantly higher when compared to the BCCAL rats exposed to saline (*p* = 0.009) and LPS (*p* = 0.009), respectively. Otherwise, the KYNA levels in the hippocampal and cerebral cortex were not significantly different between groups. The neurotoxic branch of the KP was highly activated in the sham + LPS rats, evidenced by a significantly lowered hippocampal KYNA/QA ratio compared to the BCCAL + saline (*p* = 0.015) and BCCAL + LPS (*p* = 0.011) rats, respectively, F(2,15) = 10.580, *p* = 0.001 (one-way ANOVA) (Figure 8). However, the hippocampal QA and KYNA/QA ratio levels did not significantly differ between the BCCAL rats treated with LPS and saline, respectively. These findings indicate that heightened IDO levels in LPS-treated CCH rats were associated with a reduction in neurotoxic downstream metabolites of the KP at 72 h post-LPS.

## 3. Discussion

In this study, we unveiled two major findings. First, we demonstrated that the rats in the CCH state exhibited deprivation in natural and learned behaviours following treatment with a relatively mild LPS dose. These behaviour impairments were mild compared to the sham-operated rats, which exhibited robust behaviour impairments. These findings were consistent with features of delirium in older people, in which they often presented with transient and subtle signs of inattention and drowsiness [43,44]. Secondly, we demonstrated that these delirium-like features in CCH rats were associated with downregulated KP activity and decreased hippocampal QA. In this part, we suggest that KP plays a role in delirium presentation in CCH rats treated with LPS through the immunomodulatory activity of the IDO and endotoxin tolerance.

In this study, we performed permanent BCCAL for an animal model of CCH, and our findings support CCH-induced neurodegenerative changes in the brain, evidenced by increased amyloid-β expression in the hippocampus. Ligation of both carotid arteries reduces approximately 30% to 60% of the cerebral blood flow (CBF) and gradually recovers up to 90% after eight weeks, depending on collateral circulations [45]. The latest study by Mun and colleagues (2023) reported that the CBF was reduced to a level similar to aged mice at eight weeks after the BCCAL [23]. CCH drives the ageing process through dysregulation in innate immune systems and is associated with heightened levels of inflammation. Ischaemic and hypoxia caused by CCH are related to acute and chronic neuroinflammation, in which the brain parts become necrotic and infarcted in an acute stage, and the neuron becomes apoptotic in a chronic stage. Additionally, our findings indicate increased IDO activity and TRP degradation towards the KP in rats with pre-existing CCH. The role of IDO in cerebrovascular pathologies has been reported in the literature. Previously, Honsi and colleagues (2008) reported time-course upregulated hippocampal IDO expression in mice models of cerebral ischaemia 72 h after the BCCAL [46]. Meanwhile, several studies identified that IDO is upregulated in acute and chronic phases of cerebral ischaemia and chronic brain damage after stroke injury [35,47,48]. Although our study did not suggest memory deficit in CCH rats, increased IDO was associated with post-stroke cognitive impairment, depression, and mortality [47,49]. In addition, studies suggest that altered IDO/KP activity could be the pathophysiological factor that worsens the neurodegenerative process induced by CCH [20,50]. Taken together, our findings indicate that BCCAL in Sprague Dawley rats result in CCH-induced neurodegeneration, and this ischemic cascade is contributed by amyloid-β deposition and increased IDO/KP activity.

Several lines of evidence support the presence of vascular risk factors in delirium. Otomo and colleagues (2013) identified the association between delirium with pre-existing cerebral ischemia in elderly patients who had undergone coronary artery bypass graft surgery (CABG) [30]. From perioperative magnetic resonance imaging (MRI), the patients with delirium had a significantly higher prevalence of multiple cerebral infarcts with no prior history of stroke. CCH is associated with a wide range of cardiovascular disorders, and these conditions are the predisposing factors for cerebral ischaemia and delirium, including atrial fibrillation, atherosclerosis, heart failure, and peripheral arterial disease [51,52,53,54]. Abawi and colleagues (2016) reported a 5% increased risk for delirium in patients undergoing transcatheter aortic valve replacement (TVAR) procedure via non-transfemoral compared to the transfemoral approach [52]. The author suggested that advanced vascular ischaemia in the non-transfemoral cohort could contribute to the higher incidence of POD after TVAR. Meanwhile, Honda and colleagues (2016) identified a 33% incidence rate of delirium in low-output heart failure and suggested the link possibly associated with cerebral hypoperfusion [53]. CCH during ageing induces brain capillary degeneration and impaired delivery of energy substrates to neuronal tissue, compromising neuronal stability [28,55]. To relate with our delirium model, superimposed LPS administration in CCH rats could further exacerbate cerebral hypoperfusion via nitrous oxide (NO)-induced cerebral vasodilation and impaired cerebral autoregulation, leading to delirium [56,57]. Although the underlying mechanism of how these disorders contribute to delirium is poorly understood, studies identified that subcortical ischaemic vascular disease has disrupted frontal subcortical circuits and attributed to poor executive functions [58,59,60]. This underlying mechanism is significant for patients with POD, as they often present with poor perioperative performance in executive function and slower cognitive processing speed.

The present study suggests a possible intrinsic mechanism in CCH rats associated with endotoxin tolerance dependent on IDO. Endotoxin tolerance refers to a refractory state of a cell to the immune challenge after initial inflammatory exposure from endotoxin LPS or tissue damage [61,62]. In the context of delirium in CCH rats, the hyperinflammatory state during CCH was followed by a compensatory hypoinflammatory phase in response to LPS treatment, characterised by endotoxin tolerance. Several studies have implicated the role of IDO in endotoxin tolerance, but the precise mechanisms are still being elucidated [63,64]. Previously, Salazar and colleagues (2017) identified that Toll-like receptor 4 (TLR4) ligation in LPS-conditioned dendritic cells induced elevated levels of IDO isoforms and AhR, with upregulated anti-inflammatory interleukin-10 (IL-10) and non-canonical NF-ƙB pathway activation [63]. Meanwhile, recurrent LPS challenge in IDO-deficient mice indicated the combined effects of AhR, IDO, and the cytokine transforming growth factor-β (TGF-β) to establish endotoxin tolerance. In LPS-treated CCH rats, heightened basal IDO is essential for maintaining immune tolerance via suppression of adaptive immunity from over-response to threats. During endotoxin tolerance, the immunosuppression activity of IDO is regulated via two main mechanisms, the effects of KYN signalling through the aryl hydrocarbon receptor (AhR) and the rapid depletion of TRP [65]. KYN acts as an endogenous ligand and a potent agonist for the AhR [66,67]. The activation of AhR leads to various immunomodulatory effects, including promoting the differentiation of Treg that can maintain immune tolerance. In addition, AhR also fosters the development of regulatory dendritic cells and inhibits the differentiation of pro-inflammatory Th17 cells that can polarize the immune cells [68]. Meanwhile, TRP depletion activates the amino acid starvation stress response through the activation of protein kinase general control nonderepressible 2 (GCN2) in cell microenvironments, leading to phosphorylation of the alpha subunit of eukaryotic initiation factor-2 (eIF-2α) [69,70]. eIF-2α plays a critical role in regulating protein synthesis, and its phosphorylation activates the integrated stress response (ISR), which helps cells adjust to adverse conditions [70]. In addition, GCN2 reduces T cell activation and promotes the differentiation of regulatory T cells (Tregs), which contributes to immunosuppression and tolerance. However, dysregulation of GCN2 and eIF-2α is associated with T cell anergy and cell death and has been implicated in various neuropathological conditions, including malignant glioma and frontotemporal dementia [71,72].

Endotoxin tolerance has been implicated in arbitrating neuroprotective effects [73,74,75]. Although the exact link between endotoxin tolerance and neuroprotection is not well established, it could be related to the immunosuppressive activities of IDO in the CNS. For instance, a study by Wang and colleagues (2020) demonstrated that IDO increased significantly in neural progenitor cells (NPC) at 3 and 8 h under oxygen-glucose deprivation and suggested increasing IDO activity over time [76]. Interestingly, inhibition of IDO attenuated NPC viability. In the meantime, Lemos and colleagues (2014) identified that systemic DNA nanoparticle (DNP) treatment in mice model of multiple sclerosis inhibited the progression and severity of experimental autoimmune encephalitis (EAE) via the STING/IFN-αβ/IDO pathway [77]. The author suggested that selective STING activation, such as DNP, could effectively treat patients with hyperimmune syndrome via the IFN-αβ stimulation and IDO-dependent T cell regulatory response. The protective effects of endotoxin tolerance outside the CNS, dependent on IDO, have been reported in the literature. Increased IDO expression in chronic periodontitis promoted the conversion of conventional T cells into Tregs in LPS and interferon-γ (IFN-γ) activated human gingival fibroblasts [78]. In gastrointestinal inflammation, Acovic and colleagues (2018) demonstrated that the presence of Tregs in an IDO-dependent manner promoted wound healing in dextran sodium sulphate (DSS) induced colitis in mice [36]. However, the effects of endotoxin tolerance in the CNS are not entirely protective. Endotoxin tolerance is an early sign of sepsis and increases the susceptibility of immune response to subsequent endotoxin exposures and subsequently increased sepsis severity [79]. In summary, our findings provide additional evidence on the immunoregulatory role of IDO/KP in the CNS, with increased IDO activity that helps to maintain immune tolerance and suppresses superimposed inflammation. Based on our interpretation of the study findings, we have developed a conceptual model of IDO/KP’s role in developing delirium in older people with pre-existing CCH (Figure 9).

The current study offers several implications in neurodegenerative disease, highlighting the role of CCH and endotoxin tolerance in the mechanistic understanding of delirium pathogenesis and promoting future research on biomarkers and translational discoveries in this complex condition. Previously, human studies advocated multifactorial risks underpinning delirium pathogenesis, contributed by precipitating factors and baseline vulnerability [80]. The current study used a clinically relevant experimental model to demonstrate delirium during the neurodegenerative process, utilising a surgically induced rat with CCH as a baseline vulnerability risk. Several putative animal models reported in the literature are based on a conceptual framework of a superimposed acute systemic inflammation-induced delirium concurrent with a pre-existing vulnerability [81]. The Network for the Investigation of Delirium: Unifying Scientists (NIDUS) recommends several key conceptual areas for using animal models in delirium research, including the validity assessment of animal models [82]. Of particular note, our model combines practical and reasonable face and construct validity, in which the model imitated the clinical features of delirium in the elderly, and the mechanism used to induce the condition replicates the currently understood disease aetiology in humans. The effects of BCCAL on cerebral hypoperfusion are gradual, mimicking the progressive nature of age-related cerebrovascular dysfunction in humans, adding some potential advantage of utilising the technique in modelling CCH and delirium.

Concerning a possible molecular basis for the manifestation of delirium during CCH, our findings serve as a platform to promote future research on IDO-induced immunological tolerance for therapeutic potential targeting CCH and neurodegenerative diseases such as delirium and dementia. Regarding CCH-induced neurodegeneration, evidence from preclinical studies demonstrated that inhibition of IDO is protective against neuroinflammation in Parkinson’s disease, Alzheimer’s disease, and depression [83,84,85]. In the meantime, delirium is common during infection and sepsis, and increased IDO activity in the early phase of sepsis is associated with endotoxin tolerance and increased sepsis severity [86,87,88]. Interestingly, we demonstrated that delirium in LPS-treated CCH rats was associated with reduced hippocampal QA, suggesting a possible neuroprotective activity from IDO activation via endotoxin tolerance. The exact role of IDO modulation and its therapeutic potential in delirium during sepsis requires further investigation.

There are several limitations in the current study. First, the rats were challenged with a relatively low amount of LPS, and we could not elicit comparable effects of IDO to KP activity. As with delirium, it was hypothesised that delirium in older people indicates a failure of the vulnerable brain to show resilience to an acute stressor. There may be limits in the ability of IDO to protect the brain cells, and the use of a higher dose of LPS may trigger an intense immune response and eventually restricts this adaptive mechanism. Nevertheless, this is yet to be proven. However, as the current study attempted to imitate the development of delirium in older people, choosing a low dose of LPS would be enough to emulate delirium-like behaviour in animals with the primed brain. It has been shown that a single systemic LPS 100 μg/kg can impair cognition in adult male rats [89]. In addition, Murray and colleagues (2012) showed that a similar single dose of LPS impaired the working memory of a mouse model of prion disease on a novel T-maze task [90]. In addition, delirium is a multifactorial condition, and several triggers have been recognised to trigger delirium in the elderly, including the use of medications, immobilisation, and electrolyte imbalances [91]. LPS administration mainly emulates the acute inflammatory response seen in infection and may not be able to capture the heterogeneity and complexity of delirium in humans. Other than that, the neuroinflammatory response to LPS may vary between animal species, contributed by differences in the immune system that can limit the translatability of our findings to humans. Secondly, we modelled LPS-induced delirium-like behaviour in rats with underlying CCH to replicate an ageing animal. Ageing is a complex process, and physical decline is not the only change that describes ageing. CCH is known to model neurodegenerative-related disorders in animal models, although some studies argued that BCCAL could trigger CCH [25]. In clinical practice, CCH is rather associated with small vessel ischaemia and large vessel atherosclerosis. Nevertheless, BCCAL is associated with increased IDO, similarly seen after haemorrhagic and ischaemic stroke, and not specifically during the ageing process or age-related conditions [92]. An injured brain from a stroke can adapt to subsequent stressor conditions through natural neuroprotective strategies in post-injury environments. Although there is limited clinical proof, in vivo and in vitro studies suggest that the previously injured brain is resilient to subsequent stress as a form of adaptation to the post-injury environment [93]. These adoptions are meant to allow for neurogenesis and the re-formation of normal functional connectivity.

## 4. Materials and Methods

### 4.1. Animals

Experiments were conducted on 6–8 months old Sprague Dawley rats (n = 42), male gender, with body weight in the range of 250–350 g. The rats were obtained from the Laboratory Animal Resources Unit (LARU), Faculty of Medicine, National University of Malaysia (UKM). These rats were kept in a specific pathogen-free environment and housed in an individual room (24 × 40 × 20 cm) with automatically controlled temperature (21–25 degrees), relative humidity (45–65%), and light–dark (12–12 h) cycles. The rats were provided with free access to food and water.

### 4.2. Animal Grouping and Protocols

Twelve rats (n = 12) were randomly selected to verify the neurodegenerative effects of BCCAL, whereby they were randomly divided into two groups (n = 6/group): BCCAL and sham-operated groups. The remaining rats (n = 30) were randomly divided into three groups (n = 10/group): post-BCCAL followed by a single intraperitoneal (i.p.) sterile saline 0.9%, post-BCCAL followed by a single i.p. LPS 100 μg/kg, and sham-operated followed by a single i.p. LPS 100 μg/kg. All procedures were approved and performed following the UKM Animal Ethic Committee, with a code number (AE/FP/2021/HASHIM/24-MAR./1158-APR.-2021-MAR.-2023).

### 4.3. Bilateral Common Carotid Ligation Surgical Procedure

The procedure was performed according to the methods described in previous literature [20,21,94,95]. During the procedure, rats were anaesthetised with a mixture of i.p. Ketamine hydrochloride 100 mg/kg and xylazine hydrochloride 10 mg/kg. After shaving, the midventral cervical incision was made under an aseptic technique. The skin was excised to reveal the right and left common carotid arteries. The bilateral common carotid arteries were doubly ligated with a 4–0 silk suture, followed by wound suturing. The sham-operated control group had a similar surgical procedure without common carotid artery ligation. Post-procedure, the rats were returned to their cage at room temperature, 25 °C, until they fully recovered from anaesthesia. An additional postoperative analgesic drug (Tramadol, 5 mg/kg, intramuscular) was administered for pain control. Incisions were checked daily to ensure they were intact, not infected (heat, swelling, pain, discharge, and redness), and completely healed for at least four days. Appetite, water consumption, general body condition (physical examination), attitude, temperature, and mobility were checked daily (at least four days). Sutures were removed 7–10 days after surgery. In addition, monitoring the animal’s body weight was performed if the appetite was inadequate. All procedures were performed after an acclimatisation period of one week.

### 4.4. Lipopolysaccharide (LPS) Treatment

The setting up of an animal model for delirium with the peripheral immune challenge was conducted as described in previous studies with some modifications [90]. Lipopolysaccharides (LPS from *Salmonella enterica* serotype Minnesota L6261, Sigma Aldrich, St. Louis, MO, USA) were dissolved in normal saline and stored at 4 °C. Eight weeks after surgery, the rats were treated with i.p. LPS, 100 μg/kg or an equivalent volume of i.p. nonpyrogenic normal saline under isoflurane anaesthesia to induce systemic inflammation. The dosage of LPS mimicked a mild infection and transient sickness behaviour [90]. Rats with significant neurological deficits and visual impairments eight weeks after surgery were excluded from LPS administration due to the possibility of interference with further behavioural assessments.

### 4.5. Behavioural Tests

Briefly, three behaviour assessment methods that mimic delirium assessment in humans were utilised: the buried food test (BFT), open field test (OFT), and Y-maze test [42]. The BFT and OFT were conducted to assess the impairment in the natural behaviour, and the Y-maze test was to evaluate for learned behaviour. These assessments were conducted 0 h (before LPS treatment) and 24 h apart after LPS injection for 72 h (Figure 10).

#### 4.5.1. Buried Food Test

The BFT was conducted to evaluate consciousness, attention, motivation, and organised thinking [42]. The procedure was carried out as described in the literature with some modifications [96]. Briefly, the rat was supplied one piece of sweetened cereal two days before the test. One hour before the test, the rat was placed in the home cage for habituation. We buried one sweetened cereal pellet 0.5 cm below the surface of clean bedding (3 cm height), which was not visible to the rat. The location of the pellet was changed randomly for every test. The rat was placed in the centre of the test cage, and we measured the latency for the rat to eat the pellet. Latency was defined as the time when the rat is placed in the cage until the pellet was found and grasped in the forepaws and/or teeth. The session ended if the rat could not find the pellet within 5 min. The apparatus was thoroughly cleaned with 70% ethanol, and the bedding was changed after the removal of each rat.

#### 4.5.2. Open Field Test

The OFT was conducted to evaluate locomotor activity, exploration, and anxiety [97]. The rat was placed in an open field chamber (40 × 40 × 40 cm) made of black acrylic under dim light. The rat was allowed to move freely for 5 min, and its movement parameters were monitored and analysed using animal tracking software (HVS Image version 2018 [HVS Image, Buckingham, UK]) per 5 min interval. Total distance travelled (m) was recorded as a measure of locomotor activity, and the times spent in the centre and latency to the centre of the open field arena were measured as an indicator of anxiety-like behaviour. Freezing time was recorded (seconds) to count for fear condition. The apparatus was thoroughly cleaned with 70% ethanol after the removal of each rat.

#### 4.5.3. Y-Maze Test

The Y-maze test was conducted to evaluate short-term memory [42]. The Y maze apparatus was made of black acrylic and placed under the dim light. Each maze consisted of three arms: the start arm, the novel arm, and the other arm. The maze measured 8 × 30 × 15 cm (width × length × height), with an angle of 120 degrees between each arm. During the experiment, the Y-maze test consisted of 2 trials separated by an inter-trial interval (ITI). In the first trial (training), the rat was allowed to explore the start arm and another arm of the maze, with the novel arm blocked for 10 min. The second trial started after 2 h of ITI. In the second trial, the rat was placed in the same start arm with free access to all three arms for 5 min. A video camera was placed above the centre of the maze to capture video of the rat’s movement. The video was analysed to calculate time spent, total arm entries, and entries into the novel arm, which indicated the spatial recognition memory (learned behaviour). The start arm and other arm were randomly chosen to avoid spatial memory error or placement bias. The apparatus was thoroughly cleaned with 70% ethanol after the removal of each rat.

### 4.6. Plasma and Brain Tissue Collection

The blood sample was obtained from the orbital sinus under isoflurane anaesthesia after each session of the behavioural test. The blood was centrifuged at 1000–2000× *g* for 10 min at 4 °C to separate the clot. The resulting supernatant was apportioned into an aliquot and stored at −80 degrees until further analysis.

After the last behavioural test at 72 h post-LPS treatment, the rats were sacrificed by decapitation with guillotine under deep anaesthesia. The whole brain was removed from the skull and dissected. Tissue was then homogenised in phosphate-buffered saline (PBS) (tissue weight (g):PBS (mL) volume = 1:9). An ultrasonic cell disrupter sonicated the suspension. The homogenates were then centrifuged for 10 min at 5000× *g* at 4 °C to obtain the supernatant. The supernatant was collected and stored at −80 °C until further analysis.

### 4.7. Neurochemical Assays

#### 4.7.1. Amyloid Beta Peptide (Aβ)

The expression of amyloid-β in the cerebral cortex and hippocampus was assayed using commercially available ELISA Kits (detection range: 15.63–1000 pg/mL) according to the manufacturer’s guide (Elabscience Biotechnology Company, Beijing, China). Optimisation was performed to confirm the dilution ratio, and the optical density (OD value) of each well was measured with EnSpire™ Multimode Plate Reader by PerkinElmer (Waltham, MA, USA) at 450 nm.

#### 4.7.2. Kynurenine Metabolite Levels

The expression of KP metabolites (TRP, KYN, KYNA, and QA) in the serum, cerebral cortex, and hippocampus were assayed using commercially available ELISA Kits (detection range: KYN: 7.82–500 pmol/mL; KYNA: 31.25–2000 ng/mL; QA: 1.57–100 ng/mL; TRP: 1.57–100 μg/mL) according to manufacturer’s guide (ELK Biotechnology Company, Wuhan, China). A preliminary experiment was performed to determine the dilution ratio, and the optical density (OD value) of each well was measured with EnSpire™ Multimode Plate Reader by PerkinElmer at 450 nm.

#### 4.7.3. IDO Enzyme Levels

The expression of IDO in the cerebral cortex and hippocampus was assayed using commercially available ELISA Kits (detection range: 20–4500 ng/L) according to the manufacturer’s guide (Shanghai Korain Biotechnology Company, Shanghai, China). A preliminary experiment was performed to determine the dilution ratio, and the optical density (OD value) of each well was measured with EnSpire™ Multimode Plate Reader by PerkinElmer (Waltham, MA, USA) at 450 nm.

Bradford assay was carried out for the hippocampus and cerebral cortex sample to predict the protein concentration used in each test according to the manufacturer’s guide. A standard used for this assay was albumin fraction V from bovine serum (BSA) (Merck, Rahway, NJ, USA). Samples were mixed with dye reagent (Bio-Rad Laboratories Inc., Hercules, CA, USA) and measured with EnSpire™ Multimode Plate Reader by PerkinElmer at 595 nm.

### 4.8. Statistical Analysis

Statistical Package for the Social Sciences (SPSS) version 26.0 was used to conduct the statistical analyses. Normality tests were explored for all parameters using the Shapiro–Wilk normality test. Results are expressed as mean ± standard error of the mean (SEM). The unpaired Student’s *t*-test or two-tailed Mann–Whitney test was used to compare two sets of data (BCCAL vs. sham-operated), and the one-way Kruskal–Wallis ANOVA test followed by a Dunn post hoc test or one-way ANOVA followed by Turkey HSD post hoc test was used for the comparison of more than two sets of data (BCCAL + saline vs. BCCAL + LPS vs. sham + LPS). Friedman’s test two-way ANOVA was used to determine the different effects of treatment at the different time points in non-normally distributed data. *p*-values of less than 0.05 were considered statistically significant.

## 5. Conclusions

In conclusion, our study successfully developed an animal model of delirium in surgically induced CCH rats. We showed that LPS treatment in CCH rats attenuated KP activity and was associated with reduced hippocampal expression of QA. The data highlight the potential neuroprotective role of IDO in the pathophysiology of delirium via endotoxin tolerance. From a clinical perspective, future studies warrant clinical validation of the mechanistic understanding of endotoxin tolerance in human delirium. In the meantime, additional studies are necessary to explore the potential therapeutic role of IDO in neurodegenerative conditions, especially in delirium.

## Figures and Tables

**Figure 1 ijms-24-12248-f001:**
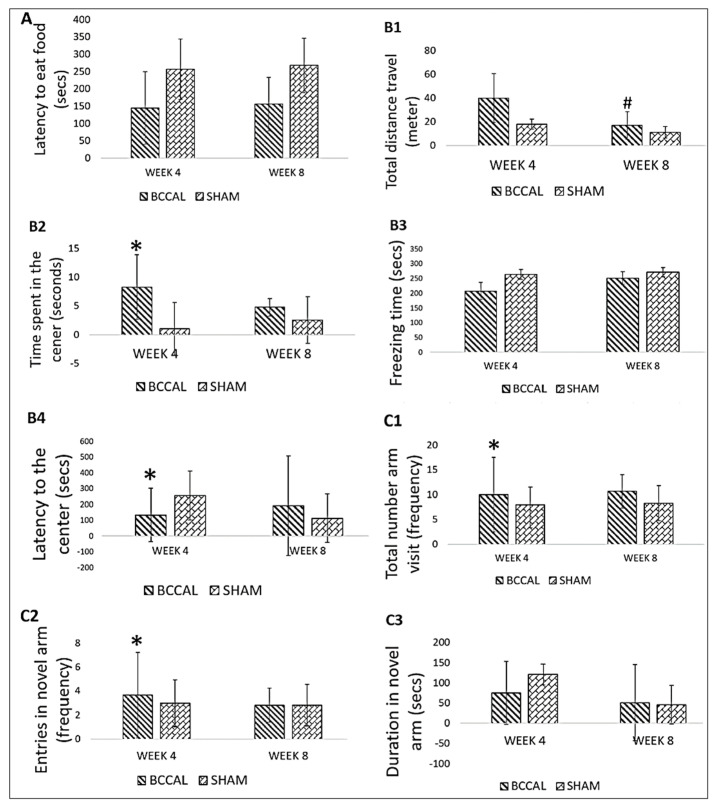
Effects of the surgery on behaviour (**A**). Latency to eat food from the buried food test between the BCCAL (n = 6) and sham-operated rats (n = 6) at four weeks and eight weeks after surgery. (**B**) Open field test, (**B1**) total distance travelled, (**B2**) time spent in the centre, (**B3**) freezing time, and (**B4**) latency to the centre between the BCCAL and sham-operated group at four weeks and eight weeks after surgery. (**C**) Y-maze test, (**C1**) total number of arms visited, (**C2**) entries to the novel arm, and (**C3**) duration in the novel arm between the BCCAL and sham-operated rats at 4 weeks and 8 weeks after surgery. * indicates *p* < 0.05 versus sham group; ^#^ indicates *p* < 0.05 versus week 4 after the surgery. Values are expressed in mean ± standard error of the mean (SEM). Abbreviation: BCCAL: bilateral common carotid artery ligation.

**Figure 2 ijms-24-12248-f002:**
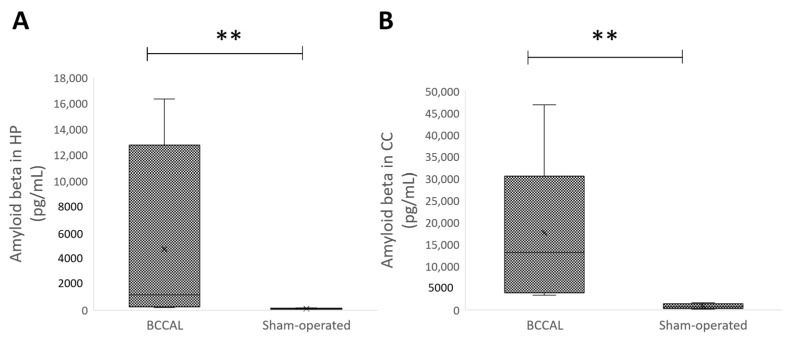
Effects of the surgery on the brain amyloid-β42 (n = 4–6). (**A**): Amyloid-β42 in the hippocampus; (**B**): amyloid-β42 in the cerebral cortex. ** indicates *p* < 0.001 versus the sham-operated rats. Values are expressed in mean ± standard error of the mean (SEM). Abbreviation: BBCAL: bilateral common carotid artery ligation.

**Figure 3 ijms-24-12248-f003:**
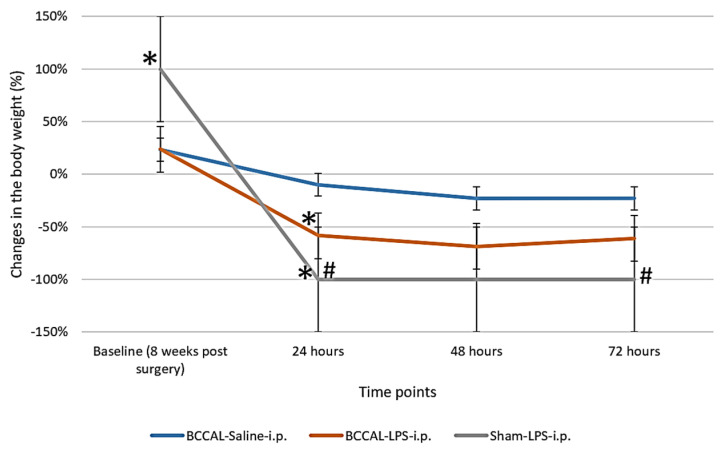
Percentage change in weight loss at the baseline (eight weeks after surgery) and 24, 48, and 72 h after LPS administration (n = 8). * indicates *p* < 0.05 vs. BCCAL-saline-i.p. and ^#^ indicates *p* < 0.05 vs. measurement at the previous time point. Values are expressed in mean ± standard error of the mean (SEM). Abbreviations: LPS: lipopolysaccharide; BCCAL: bilateral common carotid artery ligation.

**Figure 4 ijms-24-12248-f004:**
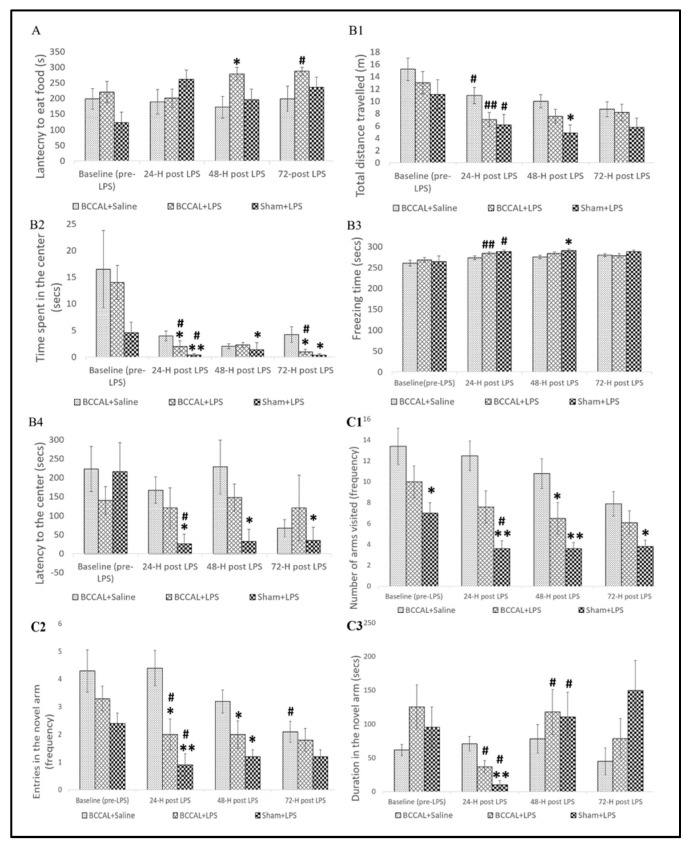
Effects of LPS on the natural and learned behaviours. (**A**) Latency to eat food from the buried food test, (**B**) open field test, (**B1**) total distance travelled, (**B2**) time spent in the centre, (**B3**) freezing time, (**B4**) latency to the centre, (**C**) Y-maze test, (**C1**) total number of arms visited, (**C2**) entries to the novel arm, and (**C3**) duration in the novel arm by surgical types and LPS administrations at 24, 48, and 72 h post-LPS (n = 10). * indicates *p* < 0.05, ** indicates *p* < 0.001 versus BCCAL + saline, and ^#^ indicates *p* < 0.05, ^##^ indicates *p* < 0.001 versus value at the previous time point. Values are expressed in mean ± standard error of the mean (SEM). Abbreviation: LPS: lipopolysaccharide; BCCAL: bilateral common carotid artery ligation.

**Figure 5 ijms-24-12248-f005:**
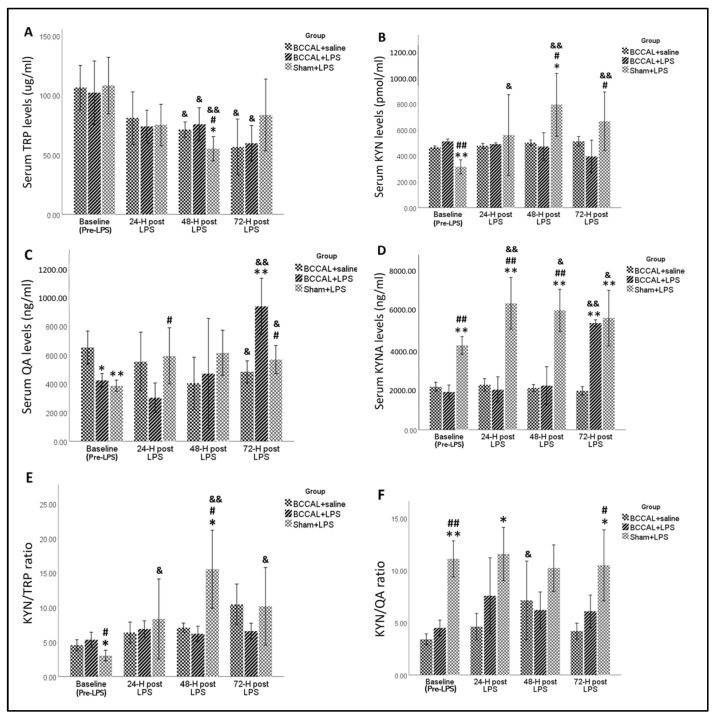
Effects of LPS on the TRP-KYN metabolites. (**A**) TRP levels, (**B**) KYN levels, (**C**) QA levels, (**D**) KYNA levels, (**E**) KYN/TRP ratio, and (**F**) KYNA/QA ratio in the serum by surgical types and LPS administrations at the baseline, 24, 48, and 72 h post-LPS (n = 6). * indicates *p* < 0.05, ** *p* < 0.001 versus BCCAL + saline, ^#^ indicates *p* < 0.05, ^##^ *p* < 0.001 versus BCCAL + LPS, ^&^ indicates *p* < 0.05, and ^&&^ indicates *p* < 0.001 versus value within the same group at the baseline. Values are expressed in mean ± standard error of the mean (SEM). Abbreviations: KYN: kynurenine; TRP: tryptophan; KYNA: kynurenic acid; QA: quinolinic acid, BCCAL: bilateral common carotid artery ligation; LPS: lipopolysaccharide.

**Figure 6 ijms-24-12248-f006:**
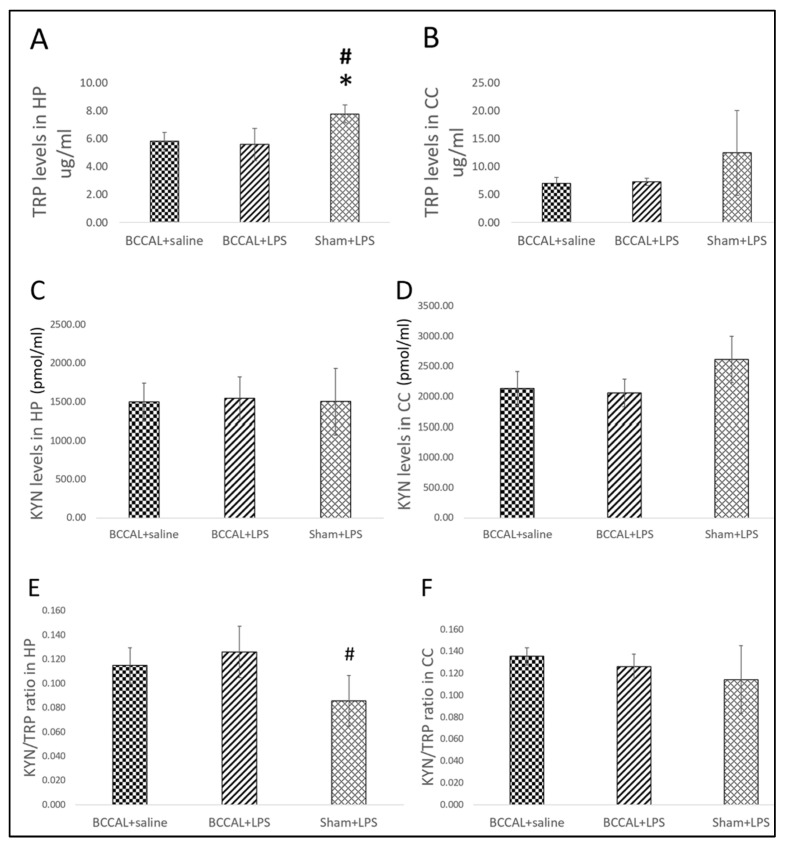
The TRP, KYN, and KYN/KYN ratio levels in the hippocampus and cerebral cortex by surgical types and LPS administrations (n = 6). (**A**) TRP levels in the hippocampus, (**B**) TRP levels in the cerebral cortex, (**C**) KYN levels in the hippocampus, (**D**) KYN levels in the cerebral cortex, (**E**) KYN/TRP ratio in the hippocampus, (**F**) KYN/TRP ratio in the cerebral cortex. * indicates *p* < 0.05 versus BCCAL + saline and ^#^ indicates *p* < 0.05 versus BCCAL + LPS. Values are expressed in mean ± standard error of the mean (SEM). Abbreviations: TRP: tryptophan; KYN: kynurenine; HP: hippocampus; CC: cerebral cortex; BCCAL: bilateral common carotid artery ligation; LPS: lipopolysaccharide.

**Figure 7 ijms-24-12248-f007:**
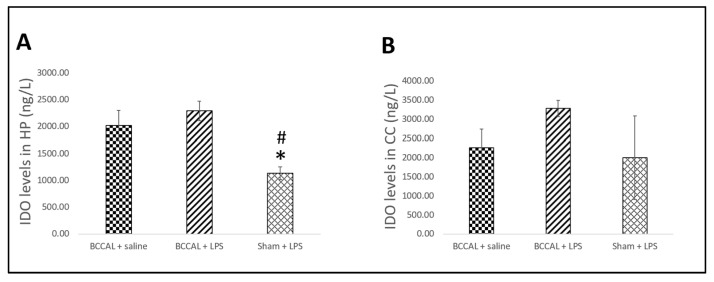
The levels of IDO enzyme in the hippocampus and cerebral cortex by surgical types and LPS administrations (n = 6). (**A**) IDO levels in the hippocampus, (**B**) IDO levels in the cerebral cortex. * indicates *p* < 0.05 versus BCCAL + saline and ^#^ indicates *p* < 0.05 versus BCCAL + LPS. Values are expressed in mean ± standard error of the mean (SEM). Abbreviations: IDO: indoleamine 2,3-deoxygenase; HP: hippocampus; CC: cerebral cortex; BCCAL: bilateral common carotid artery ligation; LPS: lipopolysaccharide.

**Figure 8 ijms-24-12248-f008:**
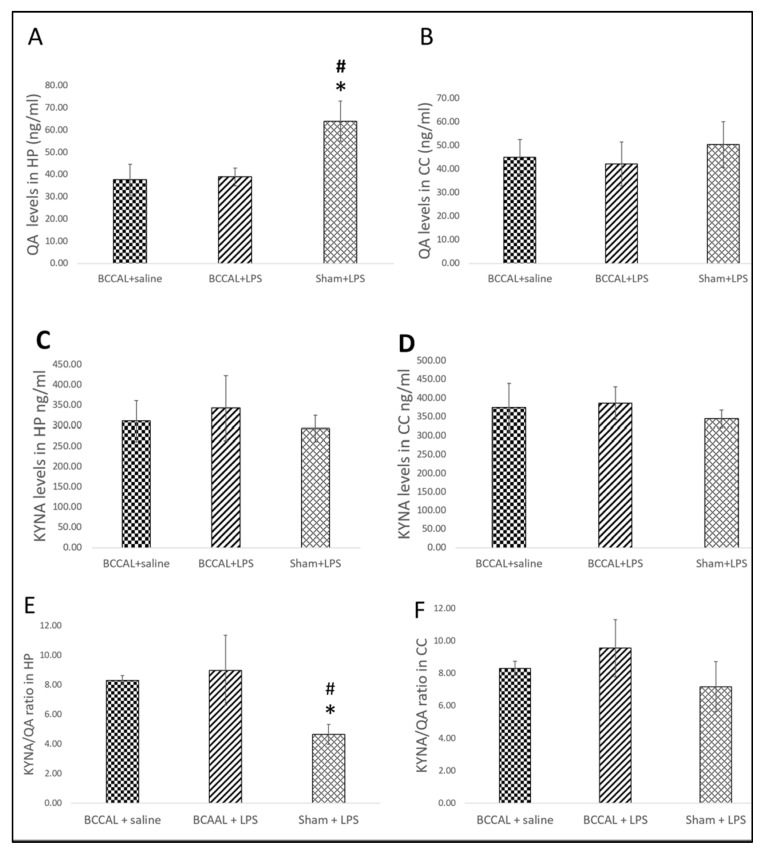
The levels of QA and KYNA in the hippocampus and cerebral cortex by surgical types and LPS administrations (n = 6). (**A**) QA levels in the hippocampus, (**B**) QA levels in the cerebral cortex, (**C**) KYNA levels in the hippocampus, (**D**) KYNA levels in the cerebral cortex, (**E**) KYNA/QA ratio in the hippocampus, (**F**) KYNA/QA ratio in the cerebral cortex. * indicates *p* < 0.05 versus BCCAL + saline and ^#^ indicates *p* < 0.05 versus BCCAL + LPS. Values are expressed in mean ± standard error of the mean (SEM). Abbreviations: QA: quinolinic acid; KYNA: kynurenic acid; HP: hippocampus; CC: cerebral cortex; BCCAL: bilateral common carotid artery ligation; LPS: lipopolysaccharide.

**Figure 9 ijms-24-12248-f009:**
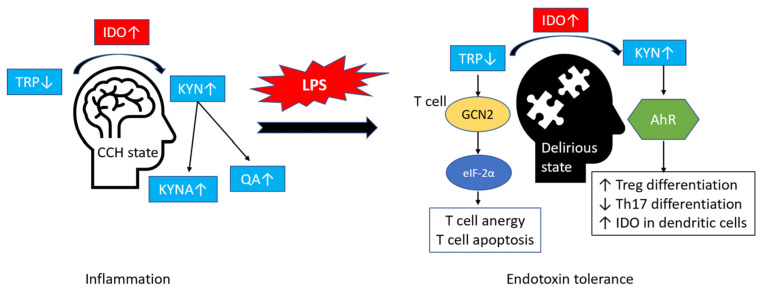
A model describing the role of IDO/KP in the development of delirium during CCH. CCH is associated with increased baseline inflammation with dysregulation of TRP-KP activity and the activation of IDO. Heightened IDO levels during CCH play a crucial role in the development of LPS-induced delirium via immunosuppressive activities and endotoxin tolerance. Arrow (
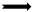
) indicates the development of delirium-like behaviour under LPS treatment in the CCH rats. Abbreviations: IDO: indoleamine 2,3-dioxygenase; TRP: tryptophan; KYN: kynurenine; CCH: chronic cerebral hypoperfusion; GCN2: general control nonderepressible 2; eIF-2α: alpha subunit of eukaryotic initiation factor-2; AhR: aryl hydrocarbon receptor; LPS: lipopolysaccharide; QA: quinolinic acid; KYNA: kynurenic acid.

**Figure 10 ijms-24-12248-f010:**
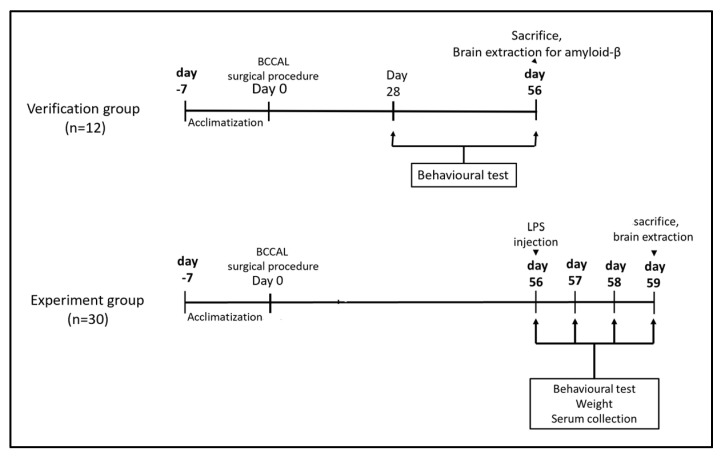
The behavioural assessment schedules in experimental rats. Abbreviations: BCCAL: bilateral common carotid artery ligation; LPS: lipopolysaccharide.

## Data Availability

The corresponding author can provide access to the dataset that was generated and/or analysed during the present study upon request, subject to reasonable conditions.

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
