# Peer review of "Lipopolysaccharide-Induced Delirium-like Behaviour in a Rat Model of Chronic Cerebral Hypoperfusion Is Associated with Increased Indoleamine 2,3-Dioxygenase Expression and Endotoxin Tolerance"

_ijms, 2023, doi:10.3390/ijms241512248_

Round 1
Reviewer 1 Report
see attached file

Reviewer 2 Report
1. The authors should provide a more detailed explanation of the methods used to induce chronic cerebral hypoperfusion in the rats. This information is crucial for replication and understanding the study's context. Additionally, it would be beneficial to reference relevant studies, highlighting the limitations and advantages of this model compared to other models.
2. It would greatly enhance the study to include a more comprehensive discussion of the potential implications of the findings in the field of neurodegenerative diseases. Specifically, how do these findings contribute to our understanding of conditions like vascular dementia and Alzheimer's disease?
3. It is important to address the limitations of using lipopolysaccharide as a model for delirium in older individuals. Are there alternative models or approaches that could provide a more accurate representation of the condition? Discuss these alternatives in detail.
4. The authors should provide a more in-depth analysis and interpretation of the role of IDO enzyme in the biological response. How does the increased IDO activity observed in the study relate to other related processes and potential therapeutic interventions?
5. The authors should acknowledge the limitations of using a rat model with underlying CCH to represent the aging process. How well does this model reflect the complexity and multifaceted nature of aging in humans?
6. Provide more information on the potential clinical implications of the study's findings. How might these findings inform future research or contribute to therapeutic approaches targeting CCH and neurodegenerative disorders?
7. The discussion section would benefit from a clearer integration of the study's major findings and their significance. Emphasize the novel aspects of the findings and their potential impact on the field.
8. In the conclusion section, summarize the key findings from the study and discuss their implications for future research.
9. Consider creating a graphical representation (e.g., figure, diagram) to visually illustrate the key concepts or relationships discussed in the manuscript.
Round 2
Reviewer 1 Report
The authors addressed most of the previous comments.
Minor comments:
1. Page 6 line 220: please specify the time point.
2. Legend of Figure 5: there may be a mistake in the legend for the && symbols (the reference to KYN/TRP ratio is not clear)
3. Page 12, line 380: not clear what 'less toxic' is compared to.
4. It would be helpful to add a brief explanation about how BCCAL + LPS best mimics the functional aspects of the disease, as compared to the sham + LPS groups, given the small differences observed in the behavioral tasks.
Only minor editing required
